# Ultrafast Photocarrier Dynamics in Vertically Aligned SnS_2_ Nanoflakes Probing with Transient Terahertz Spectroscopy

**DOI:** 10.3390/nano13010005

**Published:** 2022-12-20

**Authors:** Wenjie Zhang, Kaiwen Sun, Peng Suo, Xiaona Yan, Xian Lin, Zuanming Jin, Guohong Ma

**Affiliations:** Department of Physics, Shanghai University, Shanghai 200444, China

**Keywords:** SnS_2_ nanoflake, terahertz, edge site, ultrafast spectroscopy

## Abstract

By employing optical pump Terahertz (THz) probe spectroscopy, ultrafast photocarrier dynamics of a two-dimensional (2D) semiconductor, SnS_2_ nanoflake film, has been investigated systematically at room temperature. The dynamics of photoexcitation is strongly related to the density of edge sites and defects in the SnS_2_ nanoflakes, which is controllable by adjusting the height of vertically aligned SnS_2_ during chemical vapor deposition growth. After photoexcitation at 400 nm, the transient THz photoconductivity response of the films can be well fitted with bi-exponential decay function. The fast and slow processes are shorter in the thinner film than in the thicker sample, and both components are independent on the pump fluence. Hereby, we propose that edge-site trapping as well as defect-assisted electron-hole recombination are responsible for the fast and slow decay progress, respectively. Our experimental results demonstrate that the edge sites and defects in SnS_2_ nanoflakes play a dominant role in photocarrier relaxation, which is crucial in understanding the photoelectrochemical performance of SnS_2_ nanoflakes.

## 1. Introduction

2D materials, such as graphene, black phosphorus (BP) and transition metal dichalcogenides (TMDs) consist of single layer held together by weak van der Waals interactions, which have been paid extensive attention due to their unique mechanical, electric and optical properties [1,2,3,4]. The 2D materials also provide a playground for testing new concept in physics and are building block for designing and developing novel devices with desired functionality [5,6,7]. For instance, some TMDs have been demonstrated to have relatively high charge-carrier mobility combined with high on-off current ratios in field-effect transistors, which show potential applications in sensitive photodetectors [8,9,10], and some TMDs have exceptionally high optical absorption; these are promising applications in photovoltaic devices, etc. [11,12,13,14]. Furthermore, other layered van der Waals materials comprising earth-abundant constituents may also exhibit novel physics and applications [15,16].

Tin disulfide (SnS_2_) is a semiconducting, layered metal dichalcogenide, notably a group IV element (Sn) replacing the transition metal in TMDs. The intrinsic intermediate band-gap in the range of ~2.0–2.3 eV makes SnS_2_ suitable for photodetectors and photovoltaic devices [17,18]. As illustrated schematically in Figure 1a, the center-to-center distance between the S-Sn-S layers is 0.62 nm, and the atom layers are bonded by van der Waals interactions [19], and the fundamental band-gap is weakly effected and remains indirect in the transition from bulk to monolayer SnS_2_ [20]. As an environmentally friendly material, SnS_2_ has evoked considerable attention in applications because of its great nonlinear optical response, high electron mobility, excellent chemical stability, and wide accessibility to optoelectronic devices [21,22]. Massive studies have been focused on properties for particular applications in photo-electrochemistry [23,24,25,26,27]; however, the underlying mechanisms that dominate photoelectrochemical catalysis and photovoltaics remain elusive, which strongly correlates with photocarrier dynamics. Therefore, it is of crucial importance to understand the dynamics of the photo-generated charge carriers in SnS_2_ nanomaterials, especially the role of defects on the photocarrier dynamics. Considering the low photon energy of THz radiation (1 THz~4.1 meV), optical pump and THz probe spectroscopy (OPTP) can directly sample the dynamics of free carriers and probe the photoconductive response of materials in a noncontact way. Transient THz photoconductivity (PC) spectroscopy has been demonstrated as useful for investigating photocarrier dynamics in graphene, semiconducting and semimetallic TMDs, and other materials [4,5,6,10], which allows us to probe the charge transfer and dynamical relaxation of the desired system [28].

In 2D materials, defects play a dominant role in the dynamics of photoexcitation; therefore, the controlling defects during film growth can be a promising approach to modulating the optoelectronic properties in a variety of applications [29,30,31]. For instance, defect repair has been used to improve the electron or energy transfer in a composite of ZnO and reduced graphene oxide, which can result in enhanced optical limiting properties [32,33,34]. Stehr et al. reported that zinc vacancies contributed to the two-step two-photon absorption in ZnO [32]. By using density function theory simulation, Yan et al. found that all edges in SnS_2_ nanoribbons are semiconducting rather than only metallic edges as in MoS_2_ [35], and they predicted that the semiconducting edges in SnS_2_ could lead to lower charge-recombination rates and better photocatalytic performance. In this study, we show that edge sites in vertically aligned SnS_2_ nanoflakes can be introduced during film fabrication with the chemical vapor deposition (CVD) method, and ultrafast THz spectroscopy demonstrates that the edge sites act as defect states that can trap the photocarrier efficiently, indicating the potential application in optoelectronic field.

## 2. Experimental Details

The vertical SnS_2_ nanoflakes were grown directly on sapphire substrates (typical area 2 cm × 2 cm) using CVD method in a double-temperature tubular furnace. Stannous oxide SnO (99.999%) as the Sn source and S (99.9999%) as the sulfur source were adopted. 300 sccm argon and 30 sccm hydrogen were used as the gaseous environment. SnO was in the high temperature range of 550 °C, and S was in the low temperature range of 200 °C. The sapphire substrate was placed 5 cm below the direction of SnO, and the temperature was about 500 °C. After growing for 10 min (for thin film) and 20 min (for thick film), the temperature was naturally cooled down to room temperature before the film was moved out of the furnace. The thickness of the thin and thick SnS_2_ films was about 50 nm and 30 nm, respectively. The inset in Figure 1b shows a schematic diagram of the vertical SnS_2_ nanoflakes on a sapphire substrate.

Raman spectroscopy is a powerful tool in the characterization of the thickness and crystallographic orientation of 2D materials, especially for identifying the phonon modes of the targeted materials. The Raman spectra of our SnS_2_ nanoflakes ranging from 200 cm^−1^ to 400 cm^−1^ under 532 nm excitation are shown in Figure 1b. Clearly, we only observe one Raman mode located around 311 cm^−1^, which corresponds to the A_1g_ phonon mode of SnS_2_, arising from the out-of-plane vibration of the sulfur-tin-sulfur plane of the hexagonal phase SnS_2_ [36,37,38]. The in-plane E_g_ mode with a Raman peak located around 205 cm^−1^ has not been observed in our experiment, which is probably due to the nanosize effect [39,40]. Previous study reveals that when the thickness of the SnS_2_ nanoflakes is below 120 nm, the in-plane scattering is reduced, causing the disappearance of the E_g_ mode and the intensity reduction of the A_1g_ mode [40]. On the other hand, no Raman peaks that account for the crystalline SnO_2_ can be detected in Figure 1b, implying that surface oxidation is negligible [31]. Niwase et al. reported a close relationship between Raman intensity ratio and vacancy concentration [41]. For the Raman scattering here, the higher intensity and narrower bandwidth of the A_1g_ mode in the thick SnS_2_ sample is seen as opposed to in the thin one, suggesting the defect density in the thin film is much higher than that in thick film. Figure 1c presents the UV absorption spectra of the two SnS_2_ nanoflakes. It is clear that the absorption edges of the thick sample shift to a longer wavelength in comparison with the thin one, and both of the samples process significant absorbance at 400 nm as indicated with the black dash line. Accordingly, a 400 nm pump pulse is employed in our time-resolved THz spectroscopy study for investigating the ultrafast photocarrier dynamics. And the absorbance of thick sample at 400 nm is ~2.2, which is about 8 times higher than that in the thin one (~0.25); the difference in absorbance will be taken into account during calculation of THz PC.

In order to study the photocarrier dynamics in THz frequency, a standard optical pump and THz probe spectroscopy in transmission configuration are utilized [42]. The optical pulse train is delivered from a Ti: sapphire regenerative amplifier (Spectra-Physics, Spitfire), operating at a repetition rate of 1 kHz, a pulse duration of 120 fs and a central wavelength of 800 nm. The laser beam is split into three beams. The first one is frequency-doubled with a β-BBO crystal to output a 400 nm pulse used for ultrafast pumping. The THz pulse, co-propagating with the 400 nm pump pulse, is generated by optical rectification and detected by electro-optical sampling in a pair of 1 mm thick, (110)-oriented ZnTe crystals by the other two beams. The signal is collected with a lock-in amplifier phase-locked to an optical chopper that modulates either the THz generation arm or the pump beam at a frequency of 500 Hz. The spot size of the THz beam on the sample position is 2.0 mm, whereas the spot size of the pump beam on the sample is 3.5 mm, and the large pump spot size ensures a relatively uniform photoexcited region for the THz probe [43]. The THz beam is enclosed in a box and purged with dry nitrogen to avoid water vapor absorption. We would like to mention that the THz amplitude insertion loss of the sample was about 7% on average.

## 3. Results and Discussions

In order to intuitively analyze the differences between the two samples, the surface topographies of the two films have been characterized with scanning electron microscopy (SEM), which are shown in Figure 2a and Figure 2b, respectively. The SEM images reveal the SnS_2_ films are made up of vertically aligned SnS_2_ nanoflakes. By looking at the images illustrated in Figure 2a,b, it is clear that the wall thickness and height for the thick SnS_2_ nanoflakes are larger than those of thin film. According to previous studies, the defects in the vertically aligned SnS_2_ nanoflakes appear on the boundaries of these nanoflakes [38,39,40,41]. Therefore, it is expected that the thin nanoflakes’ film processes higher defect density than that of thick film. Figure 2c,d illustrates the schematics of the photocarrier trapping process by defect states after photoexcitation: photogenerated free carriers tend to diffuse to the boundaries before the trapping process occurs. The size of nanoflakes has an important influence on the photocarrier dynamics. In the following section, we will discuss the impact of these edge-site defects on the photocarrier dynamics by analyzing the experimental results obtained from time-resolved THz spectroscopy.

By varying the delay time between the pump pulse and the THz pulse, the transient THz transmission can be mapped out through measuring the photo-induced peak absorption of THz electric field. Figure 3a shows the typical dynamic THz transmission of both thick and thin SnS_2_ nanoflakes, which shows a similar sub-picosecond decay process. For a thin film, the Tinkham equation is applicable [44,45,46]. The change in the THz transmission is related to the PC of the film via Equation (1) [45,46].
(1)$$\Delta \sigma =\frac{n_{sub}+1}{Z_{0}}\left(\frac{1}{1+\Delta T/T_{0}}-1\right)$$
where *T*_0_ is the THz transmission without the pump, and ΔT stands for the change of the THz transmission caused by the 400 nm-pump-pulse. The negative $\Delta T/T_{0}$ reveals that the optical pump leads to positive THz PC in SnS_2_ nanoflakes. For a wide band-gap semiconductor, above band-gap photoexcitation leads to the increase of carrier population in the conduction band; the positive THz PC mainly comes from the increase of freely charged carriers in both SnS_2_ films.

In order to compare the photoelectric conversion efficiency of the two samples, we take the different absorbance at 400 nm of the two films (Figure 1c) into consideration. The inset in Figure 3a plots the transient PC (Δ*σ*) with respect to the same absorbed photon numbers at 400 nm. As Δ*σ* = Δneμ with Δn, e and μ are photocarrier density, electron charge and photocarrier mobility, respectively. Obviously, larger Δσ means higher carrier mobility μ under the assumption of the identical Δn. It is obvious that the thicker film shows double the PC than the thinner one, which suggests that the larger size nanoflake has higher carrier mobility. The subsequent relaxation process is then fitted with double exponential decays, as shown with red solid lines in the insert of Figure 3a. By taking the convolution with THz pulse, the dynamic processes can be well reproduced with Equation (2) [30]:(2)$$\Delta \sigma \left(t\right)=A_{fast}\times \exp\left(\frac{w_{THz}^{2}}{\tau_{fast}{}^{2}}-\frac{t}{\tau_{fast}}\right)\left[1-\mathrm{erf}\left(\frac{w_{THz}}{\tau_{fast}}-\frac{t}{2w_{THz}}\right)\right]\;+A_{slow}\times \exp\left(\frac{w_{THz}^{2}}{\tau_{slow}{}^{2}}-\frac{t}{\tau_{slow}}\right)\left[1-\mathrm{erf}\left(\frac{w_{THz}}{\tau_{slow}}-\frac{t}{2w_{THz}}\right)\right]$$
where *t* denotes the delay time between pump and THz pulses, *τ_fast_* and *τ_slow_* denote the lifetime of the fast and slow components, and *A_fast_* and *A_slow_* are the corresponding amplitudes of the two components, respectively. $w_{THz}$ = 0.5 ps denotes the full width at half maximum of THz pulse, and erf(*t*) is error function.

Figure 3b presents the fitting time constants with respect to pump fluence at 400 nm. It is seen that the lifetimes of both fast and slow components for both films do not show noticeable change with pump fluence. The typical value of τ_fast_ in the thin film is about 1~2 ps, which is at least 3-fold faster than its counterpart in thick film. While the magnitude of τ_slow_ in the thin film is 25~32 ps, it is slightly faster than 35~45 ps in the thick film. In brief, the thinner film has a shorter lifetime than the thicker film. In addition, Figure 3c presents the maximum THz PC with respect to pump fluence for both thin and thick SnS_2_ films. Under the photoexcitation of 24 µJ/cm^2^ at 400 nm, the maximum sheet PC (Δ*σ*_max_) of the thin film is about 0.02 mS, while the Δσ_max_ is about 0.12 mS for the thicker film under identical absorbing pump photons. Normalized with the absorbing photon density, the obtained carrier mobility in thick film is 1.7 times higher than that of the thin film. In order to further elaborate on the mechanism of the ultrafast THz response, we plot the amplitude ratio of fast and slow components (*A_fast_*/*A_slow_*) with respect to pump fluences for both films, which is shown in Figure 3d. It is seen that the fast component plays a more important role than that of slow one for both films, and this ratio increases slightly with pump fluence, i.e., the higher the pump fluence, the greater the role of the fast component. It is also noted that the dynamic process is almost dominated by the fast component in the thin film, in which the slow component is almost negligible.

According to the experimental results shown in Figure 3, we assigned the fast relaxation to photocarrier trapping via defect state. As illustrated in Figure 2c,d, after photoexcitation at 400 nm, the generated photocarriers are going to diffuse to the boundaries before recombination occurs due to the carrier concentration gradient. For the vertically aligned SnS_2_ nanoflakes, the most defects exist as edge-sites, which are located on the top and bottom ends of the nanoflakes. As a result, the defect trapping process of the photocarriers is determined by the diffusion time, which dominates the fast component in transient THz dynamics. The longer diffusion time is needed in a larger SnS_2_ nanoflake although the larger flake shows slightly higher carrier mobility. Therefore, the fast process for both thin and thick films comes from the carrier diffusion to the edge-site defect states. The relatively longer relaxation in thick film (~5 ps) than that in thin film (~2 ps) is due to the different diffusion distance: the thicker film has larger flake size that shows longer diffusion distance than that of the thin film. The electron diffusion velocity is estimated to be 10^4^ m/s via 50-nm/5-ps in thick film, which is close to the diffusion rate in Si [47]. In addition, the slow component shows a time constant on 25~45 ps time scale, which is inferred to come from the electron-hole recombination assisted by defect states.

In order to investigate the physical mechanism of photocarrier dynamics further, we deal with the PC dispersion in the frequency domain obtained at several delay times. Because of the bad signal-to-noise ratio in thin film, here we only give the measurement of THz PC dispersion in the thick film. Figure 4a shows the measured transmission of the THz electric field through the unexcited sample (blue) compared to the differential transmission of THz electric field, through the excited sample at pump delay time of around t = 0 ps normalized to the unexcited sample (red). According to the the Tinkham equation [43,44], THz PC dispersion can be obtained by applying fast Fourier transformation on the time domain THz transmission, which is displayed in Figure 4b; the blue and red dots are the typical THz PC real and imaginary parts, respectively. The complex PC dispersion can be fitted with Drude–Smith (DS) model [48]:(3)$${\tilde{\sigma }}_{DS}\left(\omega \right)=\frac{\varepsilon_{0}\omega_{p}{}^{2}\tau_{s}}{1-i\omega \tau_{s}}\left(1+\frac{c}{1-i\omega \tau_{s}}\right)$$
where *ε*_0_ is the vacuum permittivity, *ω_p_* is the plasma frequency, *τ*_s_ is the momentum scattering time, and *c* represents the backscattering constant with a range from 0 to −1; *c* = 0 denotes free charge carrier without experiencing any backscattering, while *c* = −1 presents the free charge carrier undergoing complete backscattering. The magnitude of negative *c* reflects the degree of backscattering from the crystalline surfaces or grain boundaries. Here, the nanometer-scale crystalline domain size is likely the limiting factor on the charge carrier mobility. Figure 4b shows a typical THz PC dispersion measured at the delay time of 0.5 ps along with a fitting curve with the DS model. It is clear that the real and imaginary parts of the THz PC can be well reproduced with the DS model. Figure 4c presents the fitting parameters, *ω_p_*, *τ*_s_ and *c*, for THz PC collected at delay times of 0.5, 2, 10 and 40 ps, respectively. It can be observed that *ω*_p_, which has a positive correlation with carrier density, experiences a sharp decline in the fast process (<10 ps), which indicates a large number of carriers are trapped by defects during this period, and the decrease of carrier density leads to the increase of the momentum scattering time with the delay time. Alternatively, as proposed Shimakawa et al. [49,50], non-Drude conductivity may be involved in the free carrier conductivity and tunneling carrier conductivity for nanomaterials. We also try to fit the PC dispersion in Figure 4b using the so-called Series Sequence of Free and Tunneling Carrier model [48], but all attempts to fit the experimental data failed. On the other hand, the DS model can reproduce the data well, and the fitting parameters are reasonable. Therefore, we think the DS model is a suitable model for fitting our experimental data.

Based on the experimental results and discussion above, the time evolution of photocarriers is illustrated in Figure 4d schematically. The disordered surface morphology of vertical SnS_2_ nanoflakes leads to a large number of defects, while the thin sample has more defects than the thick one. After photoexcitation, the increase in THz PC is dominated by the increase in the carrier population, and the subsequent relaxation of photocarriers follows two relaxation processes, carrier trapping by defect states and electron-hole recombination assisted by the defects. The fast time constants of 2~5 ps arise from edge-site defect trapping, and the slow relaxation with time constant of 25~45 ps is attributed to the defect-state-assisted electron-hole recombination; both of the two process show pump fluence independence.

## 4. Conclusions

In summary, we have reported the ultrafast dynamics of 2D semiconducting SnS_2_ nanoflake films with different thickness using time resolved THz spectroscopy. The defect concentration in SnS_2_ film is controlled by different thicknesses of the samples. We found that the transient THz wave transmission following photoexcitation in this nanomaterial can be fitted by bi-exponential decay function. Both the fast and slow processes are shorter in the thin film, and they are independent of pump fluence. In addition, the frequency-dependent THz PC can be well fitted with the Drude–Smith model, and the fitting parameter reveals the carrier density has changed drastically during the initial decay processes. Accordingly, we proposed the time evolution in the SnS_2_ nanoflakes. The defect trapping and the defect-assisted electron-hole recombination are responsible for the fast and slow decay progress, respectively. In general, such a method can affect the defect states by adjusting the film thickness, therefore modulating the photoelectric properties of materials, which shows application potential in this kind of 2D material.

## Figures and Tables

**Figure 1 nanomaterials-13-00005-f001:**
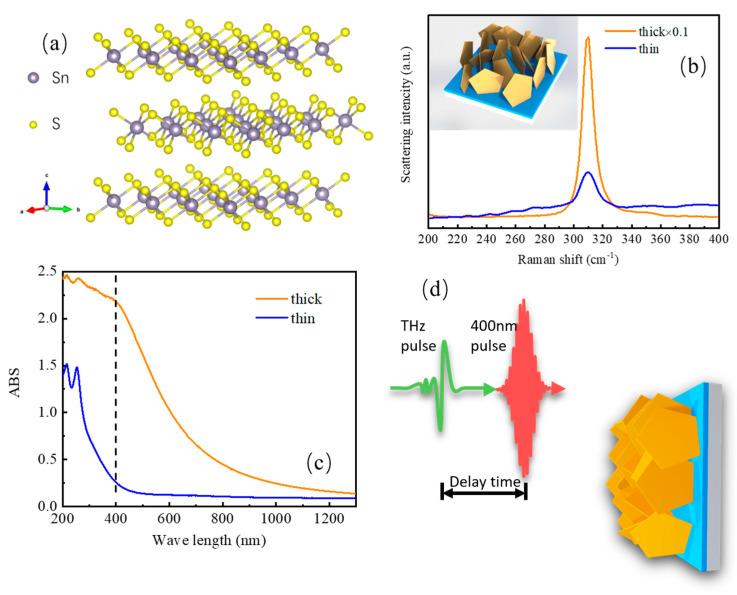
(**a**) The atomic structure of SnS_2_. (**b**) Raman spectra of SnS_2_ films with different thickness, inset shows the schematic diagram of SnS_2_ nanoflakes. (**c**) UV-visible absorption spectra of the thick and thin films of SnS_2_, respectively. (**d**) Schematic of optical-pump THz probe spectroscopy for transient THz PC measurements.

**Figure 2 nanomaterials-13-00005-f002:**
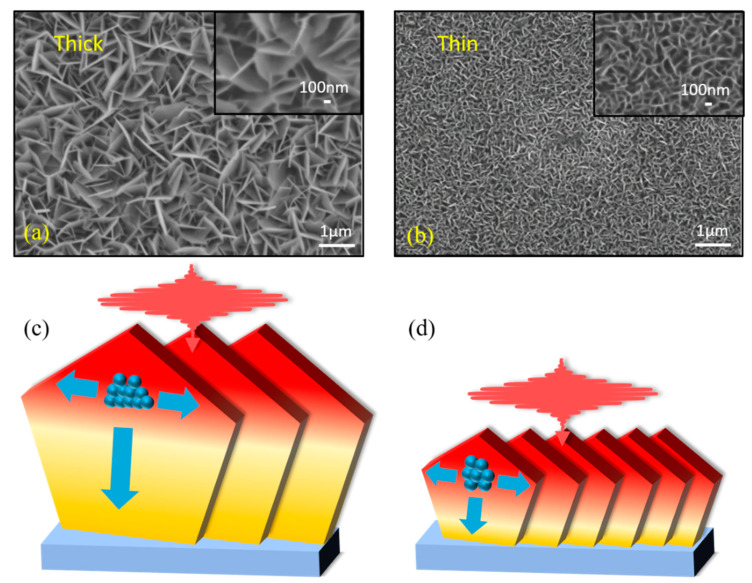
Top-view SEM images of the thick (**a**) and thin (**b**) SnS_2_ film, respectively. The schematic diagram of the sample photoexcited by the pump pulse and the photocarrier trapping in thick (**c**) and thin (**d**) SnS_2_ nanoflakes.

**Figure 3 nanomaterials-13-00005-f003:**
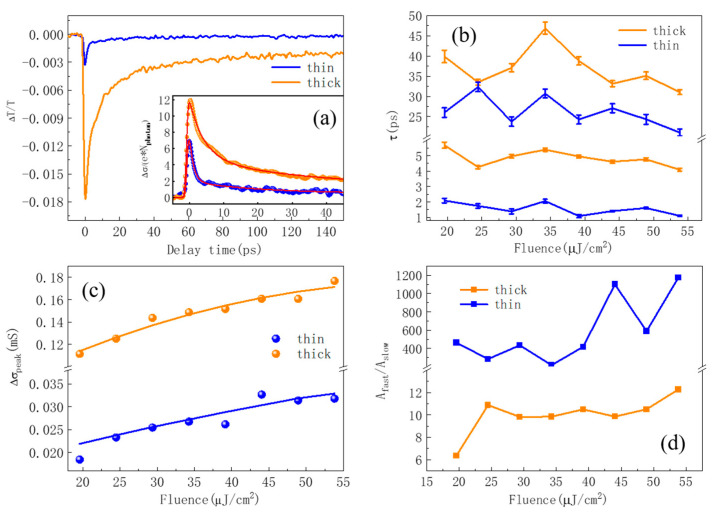
(**a**) Typical dynamic THz transmission of both thick and thin SnS_2_ nanoflakes, the inset plots the two transient THz PC together under the same absorbed photons density. (**b**) The fitting time constants (τ_fast_ and τ_slow_) with respect to pump fluence at 400 nm for both thin and thick SnS_2_ films. (**c**) The maximum THz PC with respect to pump fluence for both thin (blue) and thick (yellow) SnS_2_ films. (**d**) The amplitude ratio, i.e., A_fast_/A_slow_ with respect to pump fluence for both thin (blue) and thick (yellow) films.

**Figure 4 nanomaterials-13-00005-f004:**
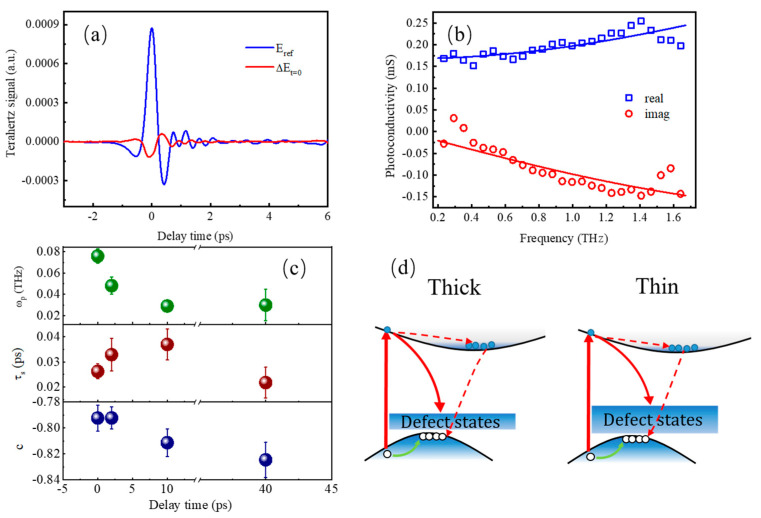
(**a**) THz signal of the unexcited SnS_2_ film (blue) and the pump-induced transmission change at time zero (red). (**b**) The real (blue square) and imaginary (red circle) parts of PC collected at delay times of 0.5 ps, in the frequency range between 0.2–1.6 THz under a photoexcitation of 400 nm with pump fluence of 24 μJ/cm^2^. (**c**) Drude–Smith fitting parameters of the PC at respective delay times. (**d**) Schematic diagram of the carrier dynamics in SnS_2_ film of different thickness.

## Data Availability

Data underlying the results presented in this paper are not publicly available at this time but may be obtained from the authors upon reasonable request.

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
