# Peer review of "Ultrafast Photocarrier Dynamics in Vertically Aligned SnS2 Nanoflakes Probing with Transient Terahertz Spectroscopy"

_nanomaterials, 2022, doi:10.3390/nano13010005_

Round 1

Reviewer 1 Report

The authors present a study of ultrafast photocarrier dynamics of vertically aligned SnS2 nanoflakes. The work is well motivated, given the growing interest on SnS2 photovoltaic devices and the surface/edge effects seen in such devices. The authors carry out pump-probe spectroscopy on the material with high-energy photoexcitation for pumping and THz spectroscopy for probing. The experimental results are modelled according to the Tinkham equation, which is based upon the Drude-Smith model. The characterizations of relaxation are particularly interesting, given the observed effects of defect states, and the interpretations seem sound. However, the authors should state the assumptions of their model, which carry forward from the Drude-Smith model. I believe that the authors' model and interpretations are valid, but some discussion of the assumptions is warranted. There is an active debate on the manifestation of Drude conductivity in the THz spectrum. For example, see [Shimakawa et al., "The origin of non-Drude terahertz conductivity in nanomaterials," Appl. Phys. Lett. 100, 132102, 2012] and the subsequent comments/citations in the literature. Beyond this request, the manuscript should be carefully proofread and (perhaps professionally) edited. There are many spelling and grammatical errors throughout the text.

Author Response

please refer to the document attached.

Reviewer 2 Report

The manuscript form Zhang et al. presents 400 nm-pump/THz-probe measurements on two different SnS2 nanoflake films. A bi-exponential decay function turned out to describe the THz dynamics, and the fast and slow relaxations are being attributed to edge-site trapping and defect assisted electron-hole recombination respectively. The manuscript presents several problematic aspects, as detailed in the following list:

1) The authors should better explain how the formula for the differential optical conductivity in (1) is derived from the Pinkham equation, or provide a reference for it.

2) The sentence at lines 155-157 (“In semiconducting SnS2 films, former process plays a more prominent role in the transient THz PC, and the contribution of increase in carrier temperature to THz PC is negligible”) also requires a Reference or a clear justification.

3) The sentence “Obviously, larger \Delta\sigma means higher carrier mobility \mu under the assumption of the identical \Delta n.“ at line 162-163 also lacks a justification. Can’t it be that \Delta n changes in the two films, due e.g. to strongly different absorption at 400 nm?

4) The formula for the double-exponential decay is not explained.  What is the role of the exp^{\omega^2/tau^2} term? What is the meaning of the \omega / \tau - t/(2\omega) argument of the error function? Using \omega to define a time (instead of a circular frequency) is extremely confusing and should be avoided. Reference 30 does not provide any information on formula (2)

5) Since both fast and slow relaxation times roughly scale with the film size, the assignment of the fast relaxation to edge states appears to be rather speculative, and should be better justified

6) If the short relaxation time is due to scattering from edge-sites one could estimate the Fermi velocity as v_F~50nm/5ps=10^4 m/s. This value is rather low compared to typical values of the Fermi velocity in metals (~10^6 m/s). Can the authors comment on that?

7) Why is the static optical conductivity not provided? This would be a useful information also to evaluate the amount of extrinsic carriers.

Author Response

please refer to the file attached

Round 2

Reviewer 2 Report

Unfortunately the authors did not provide sufficiently convincing answers to my previous report. In particular:

-point 2: I disagree with the statement that graphene at charge neutrality can be considered as a semiconductor. It is not one, nor it behaves as such. Moreover, comparing two experiments on two different materials and with different experimental parameters can not provide a convincing justification for the authors’ assumption. I may agree on the fact that carrier’s photo excitation is probably dominating the pump-probe response, but the statement that carrier’s heating is negligible is simply an hypothesis not supported by any sound argument.

point 3: I could not find in the text the statement that “The transient THz photoconductivity presented in Fig. 3a is obtained under the identical absorbed photon density at 400 nm. “

point 4: The formula discussed here was introduced in ref. 30 in order to fit the time-dependence of Kerr-rotation and not the optical conductivity. Since the differences in the optical conductivity are here attributed to the variations in the number of carriers only, formula (10) of Ref.30 should be used rather than formula (11)

Under these circumstances I can not support publication of the present manuscript

Author Response

please refer to the response letter attached

Round 3

Reviewer 2 Report

The authors have addressed the points raised by my previous review

Author Response

We have made revision on the English of the paper, also we have proofread the paper and try to correct typo, spelling and grammar mistakes. We hope this version can meet the requirement of nanomaterials.
